# Perfluorodioxolane Polymers for Gas Separation Membrane Applications

**DOI:** 10.3390/membranes10120394

**Published:** 2020-12-04

**Authors:** Yoshiyuki Okamoto, Hao-Chun Chiang, Minfeng Fang, Michele Galizia, Tim Merkel, Milad Yavari, Hien Nguyen, Haiqing Lin

**Affiliations:** 1Department of Chemical and Biomolecular Engineering, New York University, 6 MetroTech Center, Brooklyn, NY 11201, USA; hcc293@nyu.edu (H.-C.C.); fangmfeng@gmail.com (M.F.); 2School of Chemical, Biological and Materials Engineering, University of Oklahoma, Norman, OK 73019, USA; mgalizia@ou.edu; 3Membrane Technology and Research, Inc., 39630 Eureka Drive, Newark, CA 94560, USA; tim.merkel@mtrinc.com; 4Department of Chemical and Biological Engineering, University of Buffalo, The State University of New York, Buffalo, NY 14260, USA; miladyav@buffalo.edu (M.Y.); hn26@buffalo.edu (H.N.); haiqingl@buffalo.edu (H.L.)

**Keywords:** amorphous perfluorodioxolane polymers, gas separation, membrane

## Abstract

Since the discovery of polytetrafluoroethylene (PTFE) in 1938, fluorinated polymers have drawn attention in the chemical and pharmaceutical field, as well as in optical and microelectronics applications. The reasons for this attention are their high thermal and oxidative stability, excellent chemical resistance, superior electrical insulating ability, and optical transmission properties. Despite their unprecedented combination of desirable attributes, PTFE and copolymers of tetrafluoroethylene (TFE) with hexafluoropropylene and perfluoropropylvinylether are crystalline and exhibit poor solubility in solvents, which makes their processability very challenging. Since the 1980s, several classes of solvent-soluble amorphous perfluorinated polymers showing even better optical and gas transport properties were developed and commercialized. Amorphous perfluoropolymers exhibit, however, moderate selectivity in gas and liquid separations. Recently, we have synthesized various new perfluorodioxolane polymers which are amorphous, soluble, chemically and thermally stable, while exhibiting much enhanced selectivity. In this article, we review state-of-the-art and recent progress in these perfluorodioxolane polymers for gas separation membrane applications.

## 1. Introduction

Fluorinated polymers have been used extensively due to their high thermal and oxidative stability, excellent chemical resistance, superior electrical insulating ability, and unique optical properties [1,2,3]. Polytetrafluoroethylene (PTFE) and its copolymers with hexafluoropropylene and perfluoropropylvinyl compounds are crystalline and have poor optical transparency and solubility. Since 1989, several classes of amorphous perfluorinated polymers with extraordinary optical and gas transport properties were developed and commercialized by Dupont (as Teflon^TM^ AF), Solvay (as Hyflon^®^ AD), and Asahi Glass (as Cytop^TM^) [4,5,6].

Over the past 30 years, gas transport properties of these amorphous perfluoropolymers have also been investigated. These perfluoropolymers usually have high gas permeability, significant gas selectivities for some gas pairs (such as He/CO_2_ and He/CH_4_), and resistance to hydrocarbon-induced plasticization [7,8,9,10,11,12,13,14,15,16,17]. They also exhibit better resistance to thin-film physical aging than hydrocarbon polymers [18,19]. Due to these attributes, they have been used in membrane separation applications for some gases.

Various new perfluorodioxolane polymers have been synthesized over the past two decades. These perfluoropolymers are not only amorphous and soluble in fluorinated solvents, but also form thin and continuous films. Recently, we have discovered that these perfluorodioxolane films also show extraordinary gas transport properties and low swelling behavior in the presence of vapors and liquids, which make them appealing as gas separation membrane materials [20,21,22,23,24,25,26,27,28,29].

## 2. Synthesis and Physical-Chemical Properties of Perfluoropolymers

Teflon AF and Hyflon AD are a family of copolymers of tetrafluoroethylene (TFE) with (I) 2,2-bis(trifluoromethyl)-4,5-difluoro-1,3-dioxole and (II) 2,2-bis(trifluoromethyl)-4-fluoro-5-trifluoromethoxy-1,3-dioxole, respectively (Figure 1). The homopolymers of (I) and (II) are difficult to process as evidenced by their high glass transition temperature (T_g_). As a result, copolymerization of these monomers with other olefins is used to improve the processability. In the case of copolymers of TFE with (I) and (II), T_g_ decreases drastically as the molar percentage of TFE increases. For example, T_g_s of Teflon AF 1600 containing 65 mol% dioxole (I) and Teflon AF 2400 containing 87 mol% dioxole become 160 °C and 240 °C, respectively. The T_g_s of Hyflon AD 60 containing 60 mol% dioxole (II) and Hyflon AD 80 containing 85 mol% dioxole are 100 °C and 135 °C, respectively, while the T_g_ of homopolymer (II) reaches 170 °C [8].

In addition, Cytop is a homopolymer obtained by cyclopolymerization of a perfluorodiene, perfluoro-(4-vinyloxy-1-butene) specifically. Cytop is formed by penta- and hexa-cyclic structures, and the five-membered ring structure is predominant [8] (Figure 2).

Table 1 summarizes the physical and optical properties of these commercially available perfluoropolymers. These perfluoropolymers are completely amorphous, contain no hydrogen atoms, show excellent chemical and thermal stability, and are soluble in fluorinated solvents, such as hexafluorobenzene and perfluorohexane. The dielectric constants of these fluoropolymers are low, and they are almost unaffected by humidity.

The commercial amorphous perfluoropolymers exhibit excellent properties in many aspects. However, the preparation methods of these fluorinated polymers are usually complex and expensive, and the T_g_ of Cytop is relatively low. In order to overcome these limitations, we have synthesized various perfluoro-2-methylene-1,3-dioxolanes, the structures and corresponding T_g_ of their polymers are shown in Figure 3 [22].

These perfluorodioxolane monomers were prepared by direct fluorination of the corresponding hydrocarbon monomers. Figure 4 illustrates a typical synthetic scheme for a perfluorodixolane monomer and its polymer. The monomer is polymerized using a free radical initiator, such as a perfluorobenzyl peroxide. The peroxide was decomposed at 60–80 °C by a homolysis mechanism resulting in the formation of pentafluorophenyl radicals, which initiate polymerization and end up as structural units at the polymer chain end. This polyperfluoro(2-methylene-4-methyl-1,3-dioxolane), poly(PFMMD), was also synthesized using a perfluorinated precursor (perfluoro-propyleneoxide) as shown in Figure 5 [23].

These perfluorodioxolane monomers can be copolymerized with other existed fluorovinyl monomers, as well as with each other (A-H) to form amorphous copolymers that are soluble in fluorinated solvents. The composition determines the T_g_ of these copolymers. For example, when monomers D and H in Figure 3 are copolymerized, as the relative amount of monomer D increases from 20 to 75 mol%, the T_g_ varies from 105 to 155 °C accordingly. The perfluorodioxolane monomers can also be copolymerized with other fluorovinyl monomers such as chlorotrifluoroethylene (CTFE). Figure 6 shows typical examples of the copolymers [24,27].

These copolymers are also amorphous and soluble in only fluorinated solvents, and their physical properties are summarized in Table 2. When PFMMD was copolymerized with pentafluorostyrene (PFSt), the copolymers obtained were soluble in common organic solvents such as acetone and THF (Figure 7).

## 3. Gas Transport Properties

The performance of non-porous polymers is evaluated in terms of permeability and selectivity. Permeability (*P*) is defined in Equation (1) with the pressure difference (∆*p*), thickness (l), and normalized flux (*n*) [28,30,31]:(1)$$P=\frac{nl}{∆P}$$

Permeability has units of Barrer, where 1 Barrer = 10^−10^ cm^3^ (STP) cm/(cm^2^ s cmHg). For asymmetric membranes with an unknown selective layer thickness, the gas permeation property is characterized using the term of permeance (*P*/*l*), which is expressed in units of GPU, where 1 GPU = 10^−6^ cm^3^(STP)/(cm^2^ s cmHg).

In the solution-diffusion model, permeability can be expressed as the product of the concentration-averaged diffusion coefficient (D, cm^2^/s) and the sorption coefficient (*S*, cm^3^(STP) cm^−3^ atm^−1^), which represent the kinetic and thermodynamic components of the transport process, respectively [30]:(2)$$P=\bar{D}\times S$$

The selectivity, $\alpha_{A/B}$, is defined as the ratio of the permeabilities of two permeating species, *A* and *B*:(3)$$\alpha_{A/B}=\frac{P_{A}}{P_{B}}=\frac{D_{A}}{D_{B}}\times \frac{S_{A}}{S_{B}}$$
where *D_A_/D_B_* is the diffusivity selectivity, and *S_A_/S_B_* is the solubility selectivity. Modern polymeric membrane materials are designed and engineered by tailoring the sorption and diffusion behavior for selectivity improvement [31,32].

## 4. Gas Separation Properties of Perfluorodioxolane Polymers

Two perfluorodioxolane polymers, poly(PFMD) and poly(PFMMD) (as shown in Figure 6), were fabricated into 10–50 µm thin films by solvent casting or melt pressing. The films were completely dried by heating at 10 °C below the T_g_ of the polymers in air before the gas permeation tests. Figure 8 shows the gas permeability of CH_4_, N_2_, Ar, CO_2_, H_2_, and He in these two perfluorodioxolane films as a function of feed pressure at 35 °C. Gas permeability is independent of the feed pressure, suggesting that the films were defect-free [28].

Table 3 summarizes the gas separation properties of the perfluoropolymers. Poly(PFMD) and Cytop have similar fractional free volume (FFV) and T_g_. However, poly(PFMD) is less permeable (except for He) and much more selective than Cytop. Similarly, poly(PFMMD) has lower N_2_ and CO_2_ permeability than Hyflon AD 80 and much higher selectivity, while their FFV and T_g_ are similar. Gas solubility of these perfluorodioxolane polymers is generally low and depends on the applied pressure, as shown in Figure 9 [28].

Poly(PFMMD) was directly compared with its analog hydrocarbon polymer, poly(4-methyl-2-methylene-1,3-dioxolane) (poly(MMD)) [16]. Poly(PFMMD) with an FFV of 0.23 exhibits much higher gas permeability than poly(MMD) with an FFV of 0.095. For example, the CO_2_ permeabilities of poly(PFMMD) and poly(MMD) at 35 °C are 58 Barrer and 1.3 Barrer, respectively. The bulky –CF_3_ and –CF_2_ groups hinder perfluoropolymers chain packing and result in much higher FFV, and therefore increased permeability.

Table 4 compares the solubility and diffusivity of CO_2_ and CH_4_ in the perfluorodioxolane polymers, Hyflon AD 80, and Cytop [28]. Gas diffusivity can be calculated using Equation (2), and it is usually influenced by the polymer FFV and chain rigidity. Poly(PFMD) exhibits CO_2_ diffusivity one order of magnitude lower and CO_2_/CH_4_ diffusivity ratio 80% higher than poly(PFMMD) because poly(PFMMD) has a bulky CF_3_ substituent on the dioxolane ring and thus higher T_g_ and FFV. 

Interestingly, poly(PFMD) shows lower gas diffusivity than its counterpart commercial perfluoropolymer (Cytop) despite their similar T_g_ and FFV, and poly(PFMMD) exhibits much lower diffusivity than Hyflon AD 80. On the other hand, the perfluorodioxolane polymers show higher diffusivity selectivity (stronger size-sieving ability) than the commercial perfluoropolymers with similar free volume. These results can be ascribed to the narrower pore connections and larger microcavities in perfluorodioxolane polymers compared to their commercial counterparts, as evidenced by the d-spacing values from the WAXD measurements [28].

Poly(PFMD) and poly(PFMMD) also exhibit a solubility selectivity up to 65% larger relative to commercial perfluoropolymers, especially for separations involving CO_2_. The molecular origin of this behavior, which has been investigated using the lattice fluid theory [29], lies in the more favorable CO_2_ interaction with poly(PFMD) and poly(PFMMD) than commercial perfluoropolymers. This favorable interaction stems, in turn, from the higher oxygen/carbon ratio exhibited by poly(PFMD) and poly(PFMMD) relative to Teflon AF. Specifically, the oxygen/carbon ratio, which is 2:4 for poly(PFMD), 2:5 for poly(PFMMD), and 2:5.3 for Teflon AF 2400. Polar C–O–C bonds are known to interact favorably with CO_2_ molecules, which is consistent with the solubility selectivity increasing in the order: poly(PFMD) > poly(PFMMD) > Teflon AF 2400. Based on lattice fluid modeling, the CO_2_-polymer binary interaction becomes more favorable in the order: poly(PFMD) > poly(PFMMD) > Teflon AF 2400. These results mirror the orders of the oxygen/carbon ratio values, as well as of the Hildebrand solubility parameters. Specifically, the Hildebrand solubility parameter exhibited by poly(PFMD) is larger relative to poly(PFMMD), that is, it is closer to the CO_2_ value (i.e., 21.8 MPa^0.5^).

Equally important, the polymer cohesive energy density, which is represented by the characteristic pressure, p*, in the lattice fluid framework, is the key parameter to tailor solubility selectivity, all other factors (i.e., interactional pattern, free volume) being equal. Typically, solubility selectivity increases with increasing polymer cohesive energy density [29]. This conclusion is consistent with the lattice fluid characteristic pressure, p*, increasing in the order: poly(PFMD) > poly(PFMMD) > Teflon AF. 

## 5. Gas Separation Properties of Copolymers of Perfluorodioxolane

The gas permeability of these perfluorodioxolane polymers were investigated and summarized in Table 5. PFMDD (Figure 6) has two trifluoromethyl groups on the 4′ and 5′ position with *trans* and *cis* isomers. In this case, the ratio of *trans* to *cis* isomer is 73 and 27 mol%, and the polymer obtained was completely amorphous and soluble in fluorinated solvent with a T_g_ of 165 °C. PFMD was also polymerized by a free radical initiator and the obtained polymer is semicrystalline. The T_g_ and melting point are 106 and 228 °C, respectively. The copolymers of PFDMM and PFMMD with PFMD were obtained (Figure 6). The monomer reactivity ratios are almost equal and thus the resulting copolymers form an ideal random structure.

Table 5 shows that increasing the PFMD content in the copolymers decreases the gas permeances and increases the gas/CH_4_ selectivity due to the more efficient polymer chain packing for the less hindered PFMD, which results in stronger sieving ability [24].

Table 6 summarizes the gas permeation properties of perfluorodioxolane copolymers with CTFE [25,27]. The T_g_s of the copolymers decreased as the amount of CTFE was increased. The molecular weights of the copolymers are over 200,000, and they are still amorphous and soluble in fluorinated solvents (up to 50 mol% CTFE). The flexibility and fracture due to elongation of the films increases with increasing CTFE content in the copolymers. Increasing the CTFE content in the copolymers also increases the gas selectivity, especially for some gas pairs, such as He/CH_4_ and H_2_/CH_4_. The introduction of small polar monomers, such as CTFE, increases the efficiency of chain packing, leading to enhanced size sieving ability. Additionally, polychlorotrifluoroethylene (PCTFE) exhibits higher solubility parameter and, thus, higher CO_2_/gas solubility selectivity than PTFE [33], as well as better separation performance for He/H_2_ and He/CH_4_.

Gas transport properties of various polystyrene polymers have also been investigated. The permeability of gases such as CO_2_, CH_4_, and N_2_ is much higher in pentafluorostyrene (PFSt) than in other substituted polystyrenes [34]. Pentafluorostyrene is soluble in common organic solvents such as acetone and THF. The T_g_ is 110 °C and it is thermally stable. The copolymers of PFMMD and PFSt were prepared (Figure 7). When PFSt content is more than 20 mol% in the copolymer, the polymer was soluble in acetone, and the polymer films obtained were clear and transparent. The gas permeability and selectivity of the polymer are measured. The typical physical data for the copolymer PFMMD and PFSt are shown in Table 7 [35].

Figure 10 shows the gas permeability of the PFMMD-*co*-PFSt film as function of feed pressure and Table 8 summarizes the typical gas permeability of the film. PFMMD-*co*-PFSt exhibits much higher gas selectivity as compared with the neat PFSt polymer, but it shows lower gas selectivity than other PFMMD copolymers.

## 6. Mechanical Properties of Polyperfluorodioxolane Polymers

Polyperfluorodioxolane polymers show superior gas separation properties. However, the polymer films show some brittleness and can crack upon bending. Polymer mechanical properties can be improved by the addition of plasticizers. Perfluoropolyether (PFPE) is chemically and thermally stable and used as a lubricant in the aerospace and automotive industries. Poly(PFMMD) is relatively compatible with PFPE, and the blended films of poly(PFMMD) and PFPE are clear and transparent. Table 9 records the mechanical and gas transport properties of the blends of poly(PFMMD) and PFPE, including tensile strength (*σ*, MPa) and Young’s modulus (*E*, MPa) [37].

As the PFPE content increases from 0 to 5 wt% and 10 wt%, the breaking elongation of the blends increases from 1.3% to 4.1% and 7.36%, respectively. Young’s modulus decreases with increasing PFPE content, indicating reduced brittleness and increased flexibility. The gas permeance, especially N_2_ and O_2_, increases with increasing PFPE content, while gas selectivity decreases (Table 9). This finding is consistent with general transport behavior in plasticized glassy polymers. The plasticizer increases polymer chain motions and thus gas diffusion coefficients, particularly for large molecules. As a result, the size-sieving ability of the plasticized polymeric membrane is reduced, resulting in a decrease in selectivity.

Another way to improve the film flexibility is to copolymerize with a smaller monomer to reduce the steric crowding along the polymer chain while ensuring the amount of the smaller monomer is suitable to retain the amorphous nature. The best co-monomer choice would be TFE. However, TFE is not easy to handle in a common chemical laboratory. On the other hand, chlorotrifluoroethylene (CTFE) can be an attractive alternative because it is commercially available and safer to handle in academic laboratories than TFE. CTFE is the most widely used fluoroalkene after TFE and vinylidene fluoride (VDF) and is readily copolymerized with various vinyl monomers to yield novel copolymers [33]. We have prepared copolymers of PFMMD and CTFE (cf. Figure 6).

Table 10 summarizes the physical properties of PFMMD and its copolymers with CTFE. The copolymers are thermally stable near 300 °C under an air atmosphere. Increasing the CTFE content in the copolymers decreases the T_g_ and Young’s modulus, and increases both tensile strength and elongation at break of the copolymer. Figure 11 also depicts the stress–strain curves for these copolymers. These results validate that the brittleness of poly(PFMMD) films can be reduced, and film flexibility can be improved when CTFE is introduced in the polymer chains [25].

## 7. Relationships between Gas Permeability and Selectivity and Robeson Tradeoff Plots

For polymers used as membrane materials for gas separation, it is desirable that they have high gas permeability combined with high selectivity for one gas over another. High permeability is important to reduce the amount of membrane area required to perform a gas separation process economically (reducing the capital costs), while high selectivity improves the efficiency of the separation (decreasing operation costs). However, in practice, there exists a trade-off relationship between selectivity and permeability. For example, high selectivity polymers tend to possess low permeability and vice versa. The trade-off relationship between permeability and selectivity was firstly outlined empirically by Robeson in 1991 [38,39], and it was further theoretically explained by Freeman in 1999 [40]. Since then, research on gas separation membranes has focused on the discovery of polymer materials that exhibit separation properties near or above the upper bound line. Perfluoropolymers, whose properties were elucidated starting from the late 1990, helped re-define an updated upper bound in 2008. More importantly, the performance of perfluorodioxolane polymers exceeds that of commercial perfluoropolymers for the separation of light gases, such as He, N_2_, and CO_2_ (Figure 12, Figure 13 and Figure 14).

## 8. A relevant Application of Perfluoropolymers: Recovery of Helium from Natural Gas

Helium (He) is extensively used in many industrial, medical, and scientific fields. Helium’s main source is from natural gas. The content of He in natural gas can vary from 0.01 to 4.0 mol% [41]. Presently, the main industrial method of He production is cryogenic distillation, which is very energy consumable. Another more efficient method for recovering He is proposed by using gas separation membranes, which has attracted research in these fields for many years [36,42,43,44,45,46]. Perfluorodioxolane polymers exhibit much higher He/CH_4_ selectivity compared to commercial perfluoropolymers and seem promising. The obtained data points are shown to be above the Robeson upper bounds (Figure 12, Figure 13 and Figure 14).

Fluoropolymers and fluorinated liquids exhibit higher He/gas solubility selectivity than their hydrocarbon analogs [17,38], which is consistent with perfluoropolymers having unfavorable interaction with hydrocarbons including CH_4_. As a result, perfluoropolymers are very promising materials for He/CH_4_ separation.

## 9. Conclusions

Perfluorodioxolane polymers are highly stable both chemically and thermally. These polymers have the features of being amorphous and soluble in fluorinated solvents. They have also shown relatively high permeability and selectivity for gas pairs such as N_2_, CO_2_, H_2_, He, with CH_4_ compared with commercial perfluoropolymers, including Teflon AF, Hyflon AD, and Cytop. The gas separation properties of these polymers tend to be near or above the upper bound for a number of important gas pairs. In addition, the preparation of the perfluorodioxolane polymers is rather simple and practical, and it is affordable at large scale. For these reasons, perfluorodioxolane polymer-based membranes are currently being developed for a number of membrane gas separations including helium recovery and CO_2_ removal from natural gas resource.

## Figures and Tables

**Figure 1 membranes-10-00394-f001:**
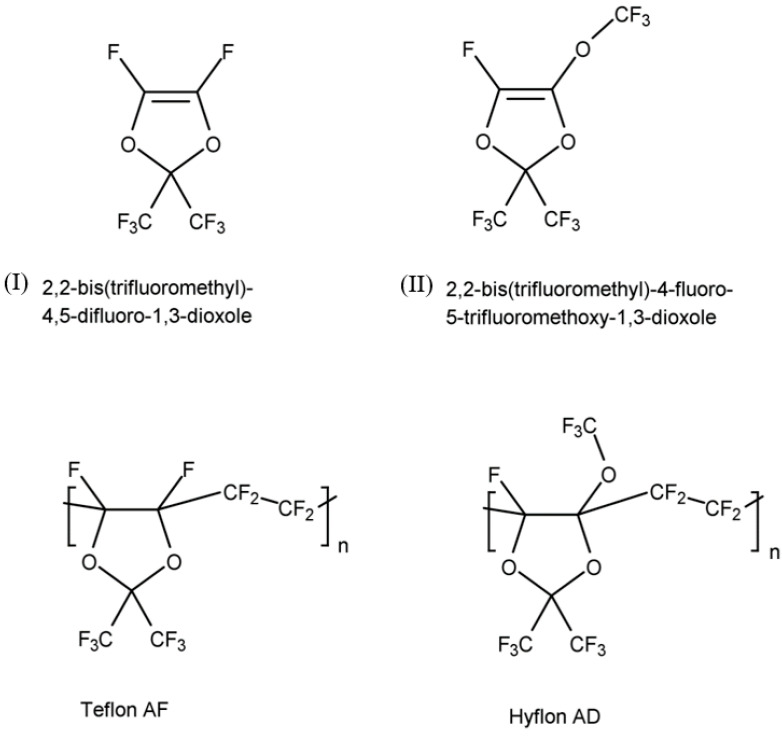
Chemical structures of monomers (I) and (II) and their copolymers with tetrafluoroethylene (TFE): Teflon AF and Hyflon AD.

**Figure 2 membranes-10-00394-f002:**
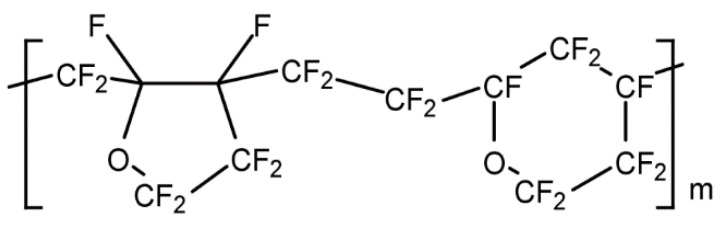
Chemical structure of the cyclic perfluoropolymer, Cytop^TM^.

**Figure 3 membranes-10-00394-f003:**
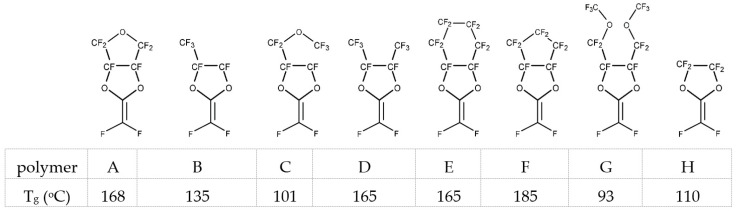
Chemical structures of perfluorodioxolane monomers and T_g_ of their polymer.

**Figure 4 membranes-10-00394-f004:**
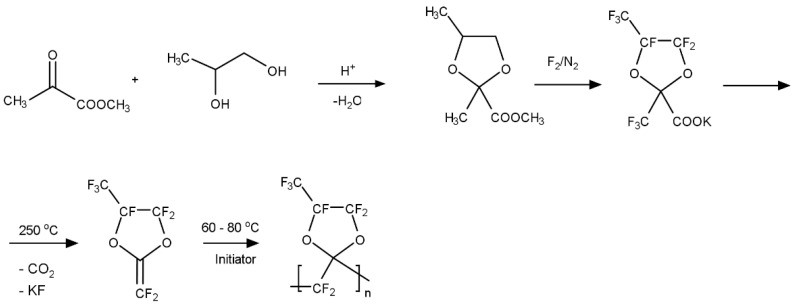
A typical synthetic route for a perfluorodioxolane monomer and the polymer. The case of poly(PFMMD) is shown.

**Figure 5 membranes-10-00394-f005:**
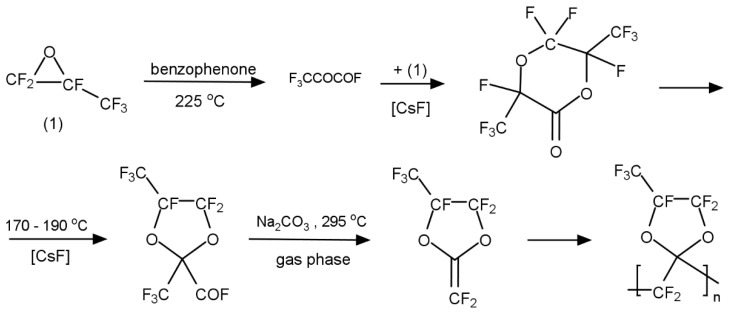
Synthesis of perfluoro-2-methylene 4-methyl 1,3-dioxolane and poly(PFMMD).

**Figure 6 membranes-10-00394-f006:**
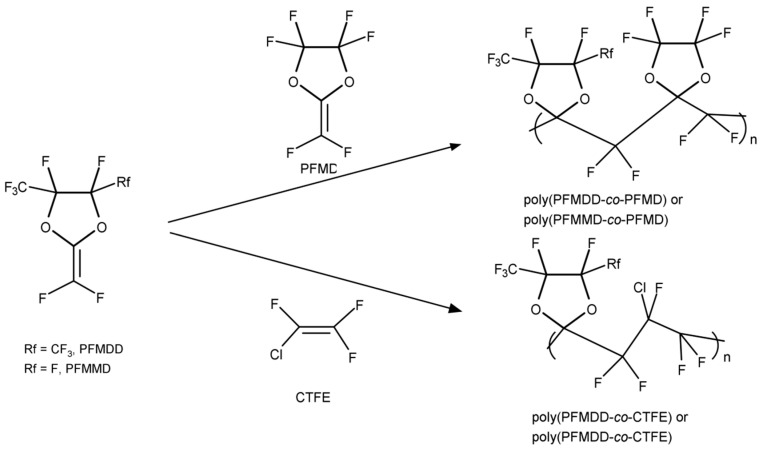
Chemical structures of the monomers for PFMDD, PFMMD, and CTFE, and the copolymers of poly(PFMDD-*co*-PFMD), poly(PFMDD-*co*-CTFE), and poly(PFMMD-*co*-CTFE) [24,27].

**Figure 7 membranes-10-00394-f007:**
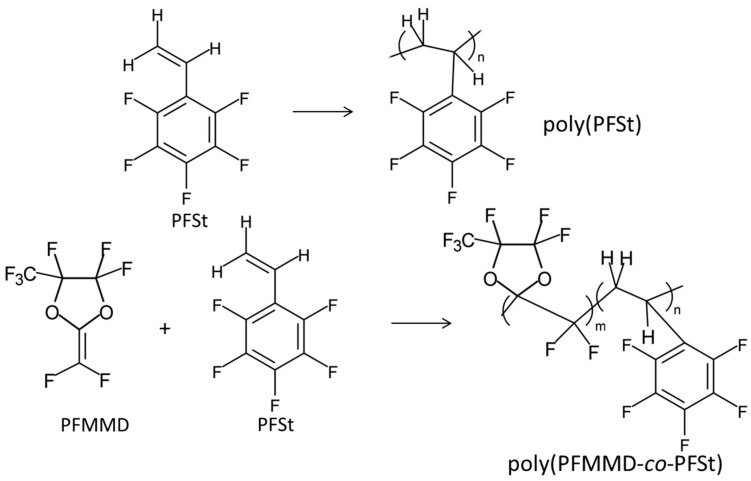
Chemical structures of polyPFSt and poly(PFMMD-*co*-PFSt).

**Figure 8 membranes-10-00394-f008:**
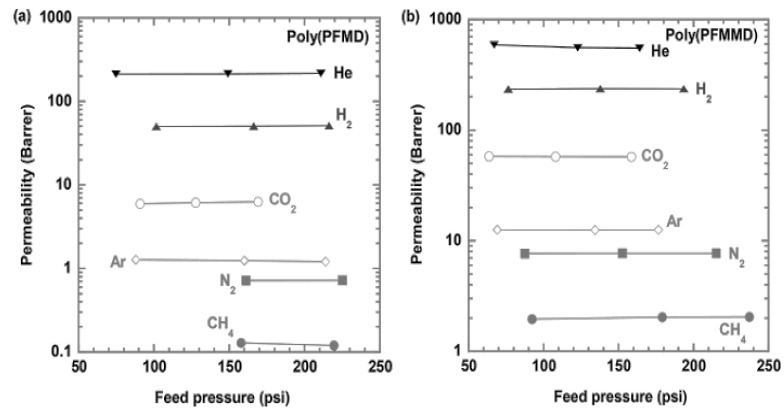
Effect of feed pressure on the pure-gas permeability of the freestanding films at 35 °C for (**a**) Poly(PFMD) and (**b**) Poly(PFMMD).

**Figure 9 membranes-10-00394-f009:**
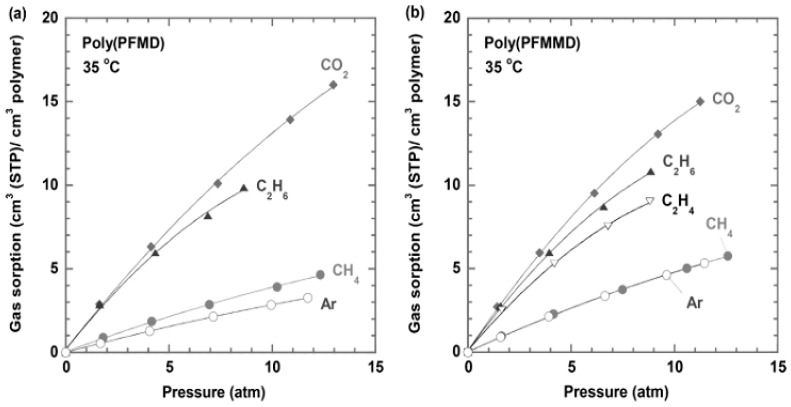
Pure-gas sorption isotherms for various gases in (**a**) poly(PFMD) and (**b**) poly(PFMMD) films at 35 °C [28].

**Figure 10 membranes-10-00394-f010:**
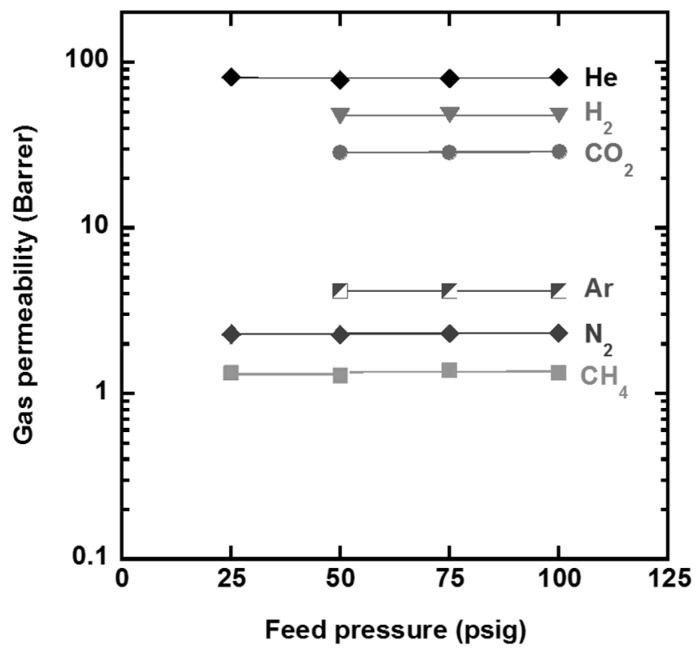
Effect of feed pressure on the pure gas permeability of the film (50 mol% PFSt).

**Figure 11 membranes-10-00394-f011:**
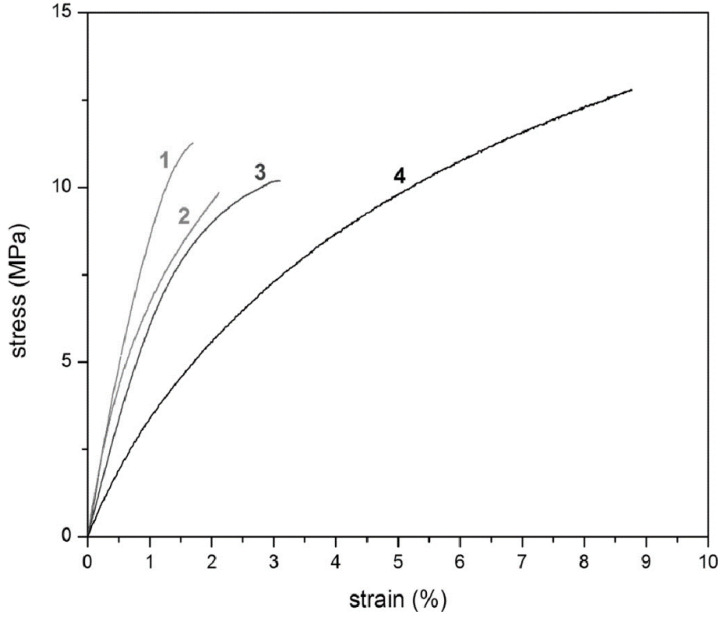
The stress–strain curves of polymer films obtained at room temperature for samples (1) poly(PFMMD), (2) poly(PFMMD-*co*-CTFE) (23 mol% CTFE), (3) poly(PFMMD-*co*-CTFE) (30 mol% CTFE), and (4) poly(PFMMD-*co*-CTFE) (50 mol% CTFE) [26].

**Figure 12 membranes-10-00394-f012:**
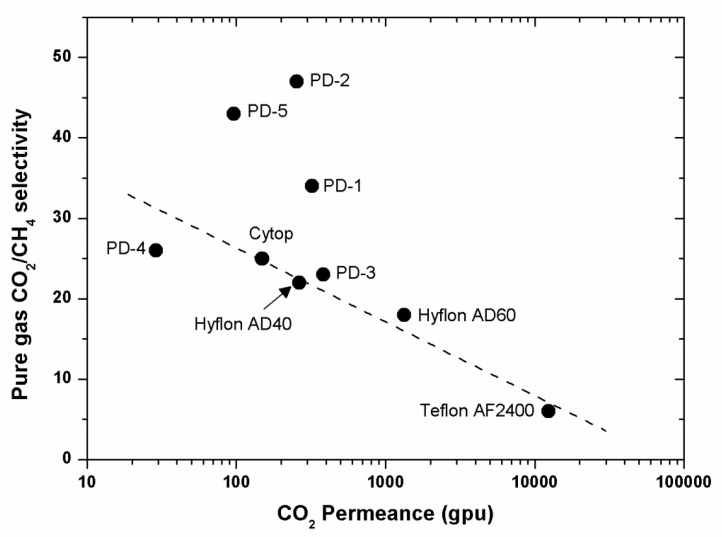
Pure-gas CO_2_/CH_4_ selectivity as a function of CO_2_ permeance for various perfluorinated polymer membranes. All data were measured at 22 °C, 50 psig feed pressure, and 0 psig permeate pressure. PD-1: PFMDD-*co*-PFMD 58:42 mol%; PD-2: PFMMD-*co*-PFMD 60:40 mol%; PD-3: poly(PFMMD); PD-4: poly(PFMD); and PD-5: poly(PFMMD-*co*-CTFE) 50:50 mol%.

**Figure 13 membranes-10-00394-f013:**
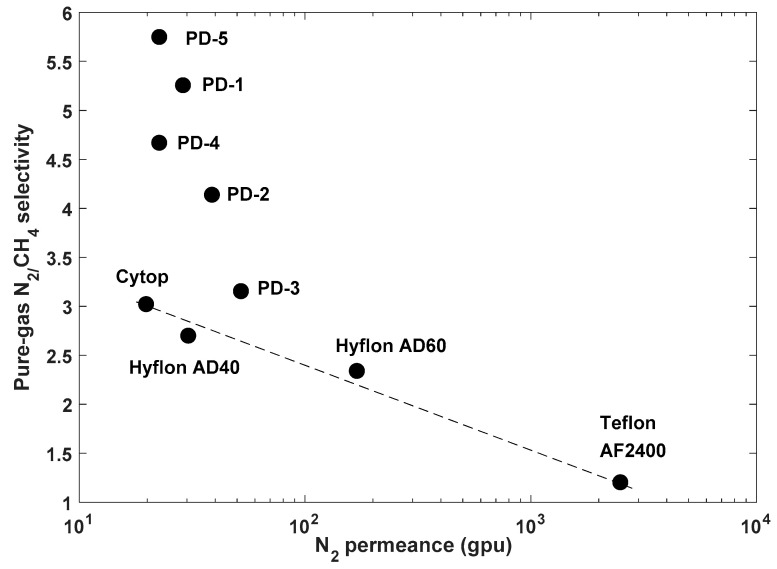
Pure-gas N_2_/CH_4_ selectivity as a function of N_2_ permeance for various perfluorinated polymer membranes. All data measured at 22 °C, 50 psig feed pressure and 0 psig permeate pressure. PD-1: PFMMD-*co*-PFMD 40:60 mol%; PD-2: PFMMD-*co*-PFMD 50:50 mol%; PD-3: PFMMD-*co*-PFMD 75:25 mol%; PD-4: poly(PFMMD-*co*-CTFE) 63:37 mol%; and PD-5: poly(PFMMD-*co*-CTFE) 50:50 mol%.

**Figure 14 membranes-10-00394-f014:**
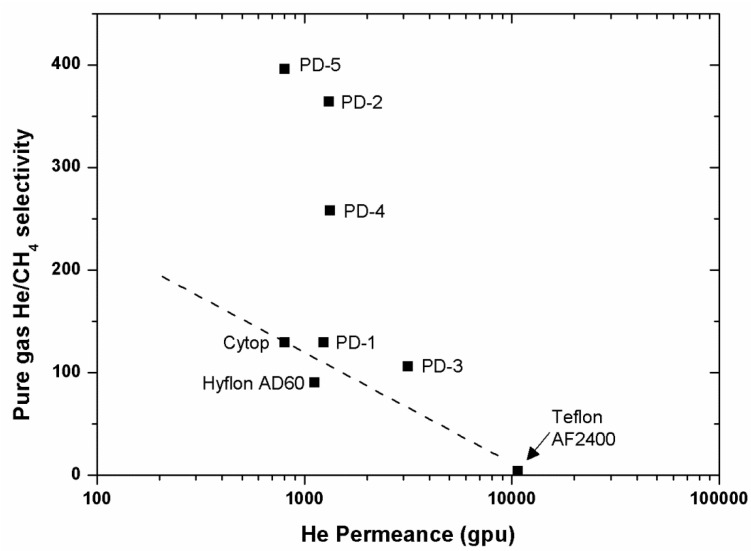
Pure-gas He/CH_4_ selectivity as a function of He permeance for various perfluorinated polymer membranes. All data were measured at 22 °C, 50 psig feed pressure, and 0 psig permeate pressure. PD-1: PFMDD-*co*-PFMD 58:42 mol%; PD-2: PFMMD-*co*-PFMD 60:40 mol%; PD-3: poly(PFMMD-*co*-PFMD) 40:60 mol%; PD-4:poly(PFMDD-co-CTFE): 43:47 mol%; and PD-5: poly(PFMMD-*co*-CTFE) 50:50 mol%.

**Table 1 membranes-10-00394-t001:** Physical Properties of Teflon AF, Hyflon AD, and Cytop.

Properties	Teflon AF		Hyflon AD		Cytop
	1600	2400	60	80	
T_g_ (°C)	160	240	100	135	108
Refractive index	1.31	1.29	1.33	1.32	1.34
Density (g/cm^3^)	1.78	1.67	1.93	1.80	1.84
Dielectric constant (ε)	1.93	1.90	1.87	2.00	2.1–2.2

**Table 2 membranes-10-00394-t002:** Physical Properties of Various Perfluorodioxolane Copolymers.

Samples	Copolymers			T_g_ (°C)
	Name	Composition (mol:mol)		
1	PFMDD-*co*-PFMD	PFMDD:PFMD =	74:26	155
2			58:42	145
3			43:57	125
4	PFMDD-*co*-CTFE	PFMDD:CTFE =	80:20	152
5			65:35	132
6	PFMMD-*co*-CTFE	PFMMD:CTFE =	63:37	113
7			51:49	99
8			43:57	89
9	PFMMD-*co*-PFSt	PFMMD:PFSt =	0:100	110
10			18:82	94
11			35:65	90

**Table 3 membranes-10-00394-t003:** Comparison of Physical Properties and Gas Separation Properties (35 °C) of Poly(PFMD), Poly(PFMMD), and Commercial Glassy Perfluoropolymers [9,28].

Polymers	T_g_ (°C)	FFV	Permeability (Barrer)				Selectivity		
			He	H_2_	N_2_	CO_2_	He/CH_4_	H_2_/CO_2_	CO_2_/CH_4_
Poly(PFMD)	111	0.21	210	50	0.71	5.9	1650	8.4	46
Poly(PFMMD)	135	0.23	560	240	7.7	58	280	4.1	29
Teflon AF 1600	162	0.31	-	550	110	520	-	1.1	6.5
Hyflon AD 80	134	0.23	430	210	24	150	36	1.4	13
Cytop	108	0.21	170	59	5.0	35	84	1.7	18

**Table 4 membranes-10-00394-t004:** Comparison of Solubility Parameter (δ_P_), Gas Solubility, and Diffusivity in Poly(PFMD), Poly(PFMMD), Cytop, and Hyflon AD80 at 10 atm and 35 °C [28].

Polymers	δ_P_ (MPa^0.5^)	*S_A_* (cm^3^(STP)/(cm^3^ atm))		$\frac{S_{CO_{2}}}{S_{CH_{4}}}$	*D_A_* (10^−7^ cm^2^/s)		$\frac{D_{CO_{2}}}{D_{CH_{4}}}$
		CO_2_	CH_4_		CO_2_	CH_4_	
Poly(PFMD)	10.3	1.3	0.39	3.3	0.36	0.026	14
Cytop		1.3	0.30	4.3	2.0	0.48	4.2
Poly(PFMMD)	10.3	1.4	0.48	2.9	3.2	0.15	9.9
Hyflon AD80	10.8	1.5	0.75	2.0	8.2	1.1	7.7

**Table 5 membranes-10-00394-t005:** Gas Separation Properties of Thin-Film Composite Membranes Based on PFMMD-*co*-PFMD and PFMDD-*co*-PFMD at 22 °C [24,27].

Copolymers		T_g_ (°C)	Permeance (GPU)		Gas/CH_4_ Selectivity			
Name	PFMD (mol%)		CH_4_	He	N_2_	H_2_	He	CO_2_
PFMMD-*co*-PFMD	0	135	20	2160	3.4	57	108	26
	25	132	18.7	2270	4.3	80	122	34
	40	126	7	2320	5.8	157	332	49
	60	123	7.34	2970	6.0	162	405	55
PFMDD-*co*-PFMD	26	155	17	1400	32	48	82	23
	42	145	11.5	1250	42	72	130	34
	80	125	8.3	1400	53	130	200	47

**Table 6 membranes-10-00394-t006:** Physical Properties of PFMMD-*co*-CTFE and PFMMD-*co*-CTFE and Gas Transport Properties of Their Thin Film Composite Membranes at 22 °C [24,27].

Copolymers		T_g_ (°C)	Permeance (GPU)				Gas/CH_4_ Selectivity			
Name	CTFE (mol%)		N_2_	H_2_	He	CO_2_	N_2_	H_2_	He	CO_2_
PFMDD-*co*-CTFE	20	152	42	580	1010	320	2.5	34	60	19
	35	132	9.8	450	610	345	6.1	210	480	48
	60	120	8.1	340	770	78	3.1	82	201	28
PFMMD-*co*-CTFE	37	113	20	633	1360	181	4.5	144	309	41
	49	99	12	457	1120	108	5	194	473	46
	57	89	5	254	804	44.3	5.5	284	900	49

**Table 7 membranes-10-00394-t007:** Physical data of PFMMD and PFSt.

PFSt mol %	T_g_ (°C)	Solubility in Acetone
100	110	Soluble
80	94	Soluble
50	90	Soluble
20	110	Soluble
0	135	Insoluble

**Table 8 membranes-10-00394-t008:** Gas transport properties of PFMMD-*co*-PFSt (50mol%) at 50 psig and 35 °C [34,36].

	Permeance (GFU)				Gas Pair Selectivity (Gas/CH_4_)				
	N_2_	H_2_	He	CO_2_	CH_4_	N_2_	H_2_	He	CO_2_
PFMMD-*co*-PFSt (50 mol%)	2.3	48	79	29	1.4	2.0	35	57	21
PFSt	5.9	65	88	50	5.1	1.1	12	16	10

**Table 9 membranes-10-00394-t009:** Physical and Gas Transport Properties of the Blends of PFMMD and PFPE [34].

PFPE Content (wt%)	T_g_ (°C)	*σ* (MPa)	*E* (MPa)	Permeance (GPU)			Selectivity			
				CH_4_	N_2_	CO_2_	N_2_/CH_4_	H_2_/O_2_	H_2_/CH_4_	CO_2_/CH_4_
0	135	12.1	8.5	20	68	520	5.7	5.0	57	26
5	131	18	6.1	26	86	621	3.3	3.9	46	24
10	125	17.4	4.2	29	91	657	3.2	3.9	42	23

PFPE (CF_2_-CF_2_O)_n_ Fomblin^®^ Z60 molecular weight 13,000, m.p. = −75 °C and and b.p. > 270 °C. The Mw of PFMMD used was 1.0 × 10^6^ g/mol.

**Table 10 membranes-10-00394-t010:** Physical Properties of PFMMD and PFMMD-*co*-CTFE Copolymers.

Samples	CTFE mol%	T_g_ (°C)	Intrinsic Viscosity (dL/g)	M_v_ ^a^ (10^6^)	*σ* (MPa)	Elongation at Break (%)	*E* (MPa)
1	0	135	0.175	1	9	1.7	6.6
2	23	125	0.167	1	10	2.1	4.7
3	30	120	0.16	0.99	10.2	3.1	3.3
4	50	99	0.153	1	13	8.7	1.5

^a^ Average molecular weight was estimated using the Mark-Houwink equation.

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
