# Peer review of "Perfluorodioxolane Polymers for Gas Separation Membrane Applications"

_membranes, 2020, doi:10.3390/membranes10120394_

Round 1

Reviewer 1 Report

This manuscript reviewed the structure, synthesis, physical properties and gas separation properties of perfluorodioxolane polymer membranes. The contents are well organized and scientifically sound. This manuscript can be accepted after a few minor changes.

The authors are suggested to provide some discussions on the thin-film physical aging properties of perfluorodioxolane membranes and make comparisons with conventional fluorinated polymer-based thin-film membranes.

In addition to He purification, are there any other promising applications for perfluorodioxolane polymer membranes?

In the introduction part, there are two places where many references were cited at once [7-17] and [20-29]. If possible, the reviewer suggests reducing the number of references cited in one place. 

There are some typos and grammar errors in the manuscript. Please review and make the necessary corrections. For instance, line 54, change "evidenced" to "evidence".

Author Response

Thank you for the comments of our manuscript (ID membrane 960595). Regarding the Reviewer 1 Comments:

-These polymers are chemically and thermally stable.

-The physical properties did not change by ageing.

-We also mentioned clearly the possible applications of these new perfluodioxolane polymers as membrane materials which  could  be applied for the He recovery and CO2 removing  form natural gas. These are mentioned in the Section 8 as well as in the conclusion of the paper.

-We have checked and corrected spelling throughout the manuscript.

Reviewer 2 Report

A big portion of the reference paper is from the authors of this review article, which causes concerns of heavy self-citation. As a review article, it should provide a balanced view of the topic. Please ensure that all of the current applications, ideas, and hypotheses are properly accounted for.

In the conclusion, the authors should throw more light on the potential research directions based on the current knowledge. This would inspire the readers to come up with future research projects.

The format of the references should be uniformed. Please proofread.

Author Response

Thank you for the comments of our manuscript (ID membrane 960595). Regarding Reviewer 1 comments:

  • We have checked and corrected spelling throughout the manuscript;
  • We have described clearly the differences of gas permeability and selectivity between our new perfluoropolymers and the commercial perfluoropolymers;
  • We have discussed the unique properties of our perfluorodioxolane polymers and commercial polymers; and
  • We mentioned possible future research projects in section 8 as well as relevant applications of our polymers.

Reviewer 3 Report

This is an interesting manuscript dealing with new types of polymers for gas separation. The paper can be published in Membranes, however the following questions must be addressed:
there is a lack of the experimental part describing procedures used for the determination of the discussed properties of membranes (e.g. gas separation properties and mechanical properties).
Authors should avoid citing the publications which are not easily available (e.g. ref. [47]).

Recommendation - minor revisions.

Author Response

Thank you for the reviewers’ comments of our manuscript (ID membrane 960595). Regarding reviewer 2 comments:

  • We have described briefly the physical properties of our polymers and cited in detail in our published papers for more experimental references.
  • The cited references are mostly easily available; only one reference 47 is a recent Ph.D thesis 2020.

Round 2

Reviewer 2 Report

The authors have addressed the issues in the first version. Overall, the quality of the manuscript has improved.